# Serum Response Factor (SRF) Drives the Transcriptional Upregulation of the MDM4 Oncogene in HCC

**DOI:** 10.3390/cancers13020199

**Published:** 2021-01-08

**Authors:** Rossella Pellegrino, Abhishek Thavamani, Diego F. Calvisi, Jan Budczies, Ariane Neumann, Robert Geffers, Jasmin Kroemer, Damaris Greule, Peter Schirmacher, Alfred Nordheim, Thomas Longerich

**Affiliations:** 1Institute of Pathology, University Hospital Heidelberg, 69120 Heidelberg, Germany; Jan.Budczies@med.uni-heidelberg.de (J.B.); arianeneumann@t-online.de (A.N.); Jasmin.Kroemer@med.uni-heidelberg.de (J.K.); Damaris.Greule@med.uni-heidelberg.de (D.G.); Peter.Schirmacher@med.uni-heidelberg.de (P.S.); Thomas.Longerich@med.uni-heidelberg.de (T.L.); 2Department for Molecular Biology, Interfaculty Institute of Cell Biology, University of Tuebingen, 72074 Tuebingen, Germany; abhishek.thavamani@hotmail.com (A.T.); alfred.nordheim@unituebingen.de (A.N.); 3Institute of Pathology, University Hospital Regensburg, 93053 Regensburg, Germany; diego.calvisi@klinik.uni-regensburg.de; 4Genome Analytics, Helmholtz Centre for Infection Research, 38124 Braunschweig, Germany; robert.geffers@helmholtz-hzi.de

**Keywords:** HCC, MDM4, ELK1, ELK4, ETS transcription factors, ERK, tumor protein p53, SRF, *MDM4* transcriptional regulation, XI-011

## Abstract

**Simple Summary:**

Hepatocellular carcinoma (HCC) represents the most common type of liver cancer and has a poor prognosis. Therefore, there is an urgent need for the identification of new therapeutic options. The mouse double minute homolog 4 (*MDM4*) gene, a known p53 inhibitor, is upregulated in most HCCs. Here, we aimed to investigate the mechanisms leading to MDM4 transcriptional upregulation and to evaluate whether therapeutic targeting of these mechanisms might represent a suitable approach for future therapy. Using human HCC cell lines, a mouse model, and human HCC cohorts, we have identified serum response factor (SRF), ETS transcription factors ELK1 and ELK4 as transcription factors (TFs) driving *MDM4* expression. Treatment of HCC cell lines with XI-011, a pharmaceutical inhibitor of *MDM4* transcription, reduced the expression of both the TFs and MDM4 and impaired tumor growth, suggesting that targeting the *MDM4* transcription may provide a rationale for future targeted therapy of HCC.

**Abstract:**

Different molecular mechanisms support the overexpression of the mouse double minute homolog 4 (MDM4), a functional p53 inhibitor, in human hepatocellular carcinoma (HCC). However, the transcription factors (TFs) leading to its transcriptional upregulation remain unknown. Following promoter and gene expression analyses, putative TFs were investigated using gene-specific siRNAs, cDNAs, luciferase reporter assays, chromatin immunoprecipitation, and XI-011 drug treatment in vitro. Additionally, MDM4 expression was investigated in *SRF-VP16^iHep^* transgenic mice. We observed a copy-number-independent upregulation of *MDM4* in human HCCs. Serum response factor (SRF), ELK1 and ELK4 were identified as TFs activating *MDM4* transcription. While SRF was constitutively detected in TF complexes at the *MDM4* promoter, presence of ELK1 and ELK4 was cell-type dependent. Furthermore, MDM4 was upregulated in SRF-VP16-driven murine liver tumors. The pharmacological inhibitor XI-011 exhibited anti-*MDM4* activity by downregulating the TFs driving *MDM4* transcription, which decreased HCC cell viability and increased apoptosis. In conclusion, SRF drives transcriptional *MDM4* upregulation in HCC, acting in concert with either ELK1 or ELK4. The transcriptional regulation of *MDM4* may be a promising target for precision oncology of human HCC, as XI-011 treatment exerts anti-*MDM4* activity independent from the *MDM4* copy number and the *p53* status.

## 1. Introduction

Liver cancer is the fifth most common type of cancer in the world and the second most frequent cause of cancer-related death, with limited therapeutic options for patients with advanced stages of the disease [1,2]. Hepatocellular carcinoma (HCC) accounts for the majority (80%) of liver cancer cases, and the underlying etiological factors are well known [1]. During the last few years, the molecular landscape of human HCC has been comprehensively characterized at genomic, transcriptomic, epigenomic, and proteomic levels [3,4,5,6,7]. The tumor suppressor *p53* represents the second most frequently mutated gene in HCC. Its mutation frequency shows a strong geographical variation, which parallels with the exposure to aflatoxin B1 and the prevalence of chronic HBV infection. In endemic regions, such as sub-Saharan Africa or East Asia, a mutation rate of up to 50% has been reported [8], while the *p53* mutation frequency is much lower in western countries, ranging from <10% to 25% [5,9,10]. P53 is regulated by a network of interacting factors, most prominently by genes of the mouse double minute (MDM) family [11]. Both MDM2 and MDM4 are inhibitors of p53. They are able to bind to the N-terminal transactivation domain of p53. While MDM4 is a potent inhibitor of p53 transcriptional activity [12,13], MDM2 mainly functions as a negative feedback regulator of p53 signaling. Transcriptionally activated by p53, MDM2 acts as an E3 ligase targeting p53 for proteasomal degradation [13,14]; of note, the formation of MDM2-MDM4 heterodimer complexes is essential for p53 polyubiquitination, while MDM2 alone marks p53 for monoubiquitination and thus does not promote complete p53 degradation [15]. In vitro and in vivo experiments highlighted oncogenic properties of both MDM family genes, but mouse models suggested MDM4 to be a more potent p53 inhibitor than MDM2 [16,17]. Dysregulation of MDM4 has been detected in different cancer types and various mechanisms may promote its upregulation [18,19]. We previously reported recurrent amplification of the *MDM4* gene locus in human HCC [3], which was recently validated by a large international consortium [7]. Furthermore, we proposed a post-transcriptional mechanism, by which activation of the EEF1A2/PI3K/AKT/mTOR signaling axis fosters the protumorigenic function of MDM4 in human HCC and showed that the MDM4 protein level is associated with the survival probability of HCC patients following liver resection [20].

The observation that increased *MDM4* expression levels can also be detected in human HCC samples with balanced *MDM4* gene locus, led us to hypothesize that transcriptional dysregulation may lead to an upregulation of *MDM4* in these cases. As the transcriptional regulation of *MDM4* remains largely elusive, we screened the putative basal *MDM4* promoter for transcription factor (TF) binding sites in silico, performed validation of the candidate TFs in vitro, and explored whether transcriptional dysregulation of *MDM4* might be a drug target for future translational studies. Moreover, we used an SRF transgenic mouse model that spontaneously develops HCC [21] to further validate our hypothesis in vivo. The data presented here highlight the role of SRF in driving *MDM4* transcription, which requires interaction with ELK1 or ELK4 in a cell context-dependent manner.

## 2. Results

### 2.1. Transcriptional Activation Contributes to MDM4 Upregulation in Human HCC

We have previously reported that genomic gains occur at the *MDM4* gene locus (1q32.1) and MDM4 mRNA and protein levels are upregulated in human HCCs compared to normal livers (NLs) [3,4]. However, microarray expression profiling data from 37 human HCCs revealed no significant difference in *MDM4* mRNA levels between HCCs with balanced (*n* = 13) or gained (*n* = 24) *MDM4* gene locus, respectively (balanced: 2.3 ± 0.4 vs. gained: 3.7 ± 0.4, *p* > 0.05; Figure 1A), suggesting that aberrant transcriptional activation may contribute to *MDM4* overexpression in HCC. Putative TFs involved in the regulation of *MDM4* transcription were identified by an in silico analysis of the *MDM4* promoter region using the MAPPER [22] and Jaspar [23] databases. As shown in Figure 1B, a high affinity SRF binding site (Jaspar score: 10.1), as well as binding sites for ELK1 and ELK4 (Jaspar score: 9.8 and 8.6, respectively), were detected in the 5′-UTR of the *MDM4* gene predicted to contain the basal promoter. Of note, the sequence identified as an ELK1 binding site (CCGGAA*G*) differed from the complementary reverse ELK4 recognition sequence (TTTCCGG) by only one nucleotide (*G*); therefore, this sequence was considered a putative ELK1/ELK4 binding site. ELK1 and ELK4 are ETS family proteins of the ternary complex factor (TCF) subfamily, which form ternary complexes with DNA-bound SRF [24].

Gene expression profiling revealed a significant positive association between the mRNA expression of all three putative TFs and *MDM4* in a human HCC cohort (*n* = 37; Figure 1C). These findings were corroborated in a second series of human HCC (*n* = 32; (Appendix A), which revealed significantly increased *MDM4*, *SRF*, *ELK1*, and *ELK4* mRNA levels in human HCCs compared to paired surrounding non-neoplastic liver tissues (SL) (Appendix A). Furthermore, survival analysis of the latter cohort showed that high mRNA levels of *MDM4*, *SRF*, and *ELK4* were associated with a lower survival probability of HCC patients following liver resection (*p* = 0.0003, *p* = 0.0386, and *p* = 0.0151, respectively; Figure 1D), while *ELK1* expression levels were of no prognostic value (*p* > 0.05; personal observation, 2020). Additionally, survival was analyzed using the TCGA data set (LIHC) following stratification of patients based on the *p53* gene status [7]. As shown in Appendix A, the overall survival of patients whose HCC showed a wild-type *p53* gene sequence was lower, but not significantly different when compared to *p53*-mutated cases. In addition, further stratification of HCC cases regarding the gene expression level of *MDM4*, *SRF*, and *ELK4* did not reveal significant differences in terms of survival probability between the individual groups (Appendix A). Notably, the positive association of the *MDM4* mRNA level with the gene expression of *SRF* and *ELK4* could be again validated in the large LIHC cohort (*p* < 0.001, personal observation, 2020).

### 2.2. SRF, ELK1, and ELK4 Regulate MDM4 Expression in HCC Cell Lines

As SRF is considered the central mediator of the immediate cellular serum response [25], fetal calf serum (FCS) was used to stimulate HCC cells. Indeed, both HepG2 and HLE cells that had been starved overnight showed significantly higher MDM4 mRNA and protein levels upon stimulation with FCS compared to the nonstimulated control cells (Figure 2A and Appendix A). This effect correlated with an increased phosphorylation of extracellular signal-related kinase 1/2 (ERK 1/2), indicating an active serum response (Appendix A).

To further corroborate our findings, *MDM4* gene expression was assessed in serum-starved HepG2 cells stimulated with FCS in combination with the PI3K inhibitor LY294002 or the ERK inhibitor LY3214996, respectively. The efficacy of PI3K and ERK inhibition was confirmed by detection of decreased AKT and ERK phosphorylation in treated cells compared to controls, respectively (Appendix A). As expected, the inhibition of the individual pathways prevented FCS-induced upregulation of *MDM4* expression (Appendix A). Also, knockdown of SRF expression by two different gene-specific siRNAs (siSRF_1/2) significantly reduced MDM4 mRNA and protein levels in HepG2 and HLE cells compared to transfection of a scrambled, nonsense siRNA (siNS) (Figure 2B,C). Similar results were obtained when ELK1 or ELK4 were targeted by gene-specific siRNAs in HCC cell lines (Figure 2D–G).

To validate the essential role of SRF in the upregulation of MDM4 following FCS stimulation, MDM4 protein levels were analyzed upon FCS stimulation in overnight serum-starved cells previously transfected with an SRF-specific siRNA and compared to siNS-transfected cells. As for ERK and PI3K pathway inhibition, FCS could not induce MDM4 expression in SRF-depleted HCC cells, while control cells were still FCS responsive (Figure 2H). Furthermore, MDM4 mRNA and protein levels were significantly increased in HuH7 cells, which showed low basal SRF levels (personal observation, 2020) when transiently transfected with an SRF-VP16 plasmid, which encodes for a full-length SRF cDNA fused to the transcriptional activation domain of the herpes simplex virus protein VP16 (Figure 2I). SRF-VP16 is able to bind to SRF target sites in promoter regions and activate transcription without additional co-factors that are physiologically required for SRF-mediated transcriptional activation (see below) [26].

### 2.3. ELK1 and ELK4 Are Co-Factors for SRF-Mediated Transcriptional Regulation of MDM4 in HCC Cells

To demonstrate that SRF, ELK1, and ELK4 indeed effectively induce the transcription of the *MDM4* gene, the specific siRNAs targeting SRF, ELK1, and ELK4, together with an MDM4 promoter construct carrying a Gaussia luciferase as reporter, were transiently transfected in HCC cells. Efficient depletion of SRF, ELK1, and ELK4 by gene-specific siRNAs (Appendix A) resulted in a significant decrease of luciferase activity compared to controls (Figure 3A–C).

While ELK1 is capable of initiating target gene transcription on its own [27], SRF requires additional cofactors to activate the transcription of its target genes. These include either members of the ternary complex factor (TCF) family of ETS domain proteins (ELK1, ELK4, NET) or the myocardin-related transcription factor (MRTF) family (MKL1 and MKL2) [24,28]. Modulation of *MRTF* gene expression did not affect *MDM4* mRNA expression (personal observation, 2020). In contrast, transient *ELK1* transfection significantly increased *MDM4* and *SRF* mRNA levels in HLE cells, which were blocked by previous siRNA-mediated knockdown of *SRF* expression (Figure 3D). Additionally, co-transfection of SRF and an inactive ELK1 mutant (ELK1 S383A) significantly lowered the induction of *MDM4* mRNA compared to a wildtype ELK1 cDNA in HuH7 cells (Figure 3E). Taken together, these data demonstrate that SRF drives the upregulation of *MDM4* in HCC cells, most likely in combination with either ELK1 or ELK4.

### 2.4. SRF and TCF Family Members Control the Activity of the MDM4 Promoter in HCC Cell Lines

To test for the physical interaction between the three TF candidates and the MDM4 promoter sequence in HCC cell lines, ChIP experiments were performed as outlined in Figure 4A. Specific binding of SRF, ELK1, and ELK4 were detected at their respective TF binding sites in the *MDM4* promoter in both HepG2 and HLE cells (Figure 4B). Comparing the enrichment of ELK1 and ELK4 at the *MDM4* promoter, ELK1 was more enriched in HLE cells, while ELK4 was the most prevalent TCF family member at the *MDM4* promoter in HepG2 cells (ELK1: 17.0 ± 1.93 (HLE) vs. 1.5 ± 0.04 (HepG2), *p* < 0.05; ELK4: 5.1 ± 0.92 (HLE) vs. 5.6 ± 0.27 (HepG2), *p* > 0.05). The specificity of TF binding to their cognate sequences was verified using control primers upstream (2.1 Kb) and downstream (1.3 Kb) from the basal promoter region (*p* < 0.05 for all the TFs analyzed).

Of note, *ELK1* expression was lower in HepG2 compared to HLE cells, while both cell lines expressed similar *ELK4* levels, likely explaining the differential pattern observed in the immunoprecipitation experiments (Appendix A). Furthermore, the binding of SRF to the *MDM4* core promoter was validated by overexpression of SRF-VP16, which resulted in a significant enrichment of SRF at the *MDM4* promoter in HuH7 cells compared to control cells (Figure 4C). To further investigate the relevance of SRF for *MDM4* gene expression in vivo, SRF-VP16 transgenic mice (*SRF-VP16^iHep^*), which express a constitutively active SRF-VP16 fusion protein in hepatocytes, were analyzed. These mice develop HCC via a premalignant nodular stage [21]. Immunohistochemistry revealed upregulation of the MDM4 protein in HCCs of *SRF-VP16^iHep^* mice compared to control mice (Figure 5). In line with the in vitro data, increased nuclear pELK1 and ELK4 protein expression were detected in SRF-VP16-induced HCCs compared to wildtype littermates (Figure 5).

Furthermore, *MDM4* mRNA expression was significantly upregulated in neoplastic lesions from *SRF-VP16^iHep^* mice compared to their corresponding controls (normal liver 0.13 ± 0.07 vs. HCC 1.51 ± 0.66 (mean ± SEM), Wilcoxon test *p* < 0.01), confirming that SRF activates *MDM4* transcription and results in an upregulation of MDM4 in vivo.

### 2.5. XI-011 Inhibits MDM4 Transcription by TF Downregulation

Recently, a drug screening study showed that XI-011, a pseudourea derivative, is a potent p53 activator and reduces MDM4 protein levels in breast as well as head and neck cancer cells [29]. Similarly, XI-011 treatment significantly reduced *MDM4* mRNA levels in a dose-dependent manner in HepG2 and HLE cells (Figure 6A). The strongest effect was recorded at concentrations of 0.5 and 1 µM XI-011 in both cell lines, respectively. Downregulation of MDM4 protein was consistently detected after 16 h of XI-011 treatment in these cell lines (Figure 6B) and similar results were obtained at every time point tested (Appendix A). Of note, MDM4 mRNA and protein levels were also decreased following XI-011 treatment in Hep3B cells with deleted *p53* alleles (Appendix A). In line with this, XI-011 significantly reduced *MDM4* promoter activity in HepG2 and HLE cells (Figure 6C). Of note, MDM2 protein levels were not affected by XI-011 treatment in HCC cell lines (Appendix A). XI-011-mediated reduction of MDM4 expression restored the p53 function in HepG2 cells, as indicated by the upregulation of p53 protein and induction of apoptosis (PARP cleavage) (Figure 6D). Additionally, reactivation of the p53-mediated transcription was confirmed by upregulation of *p21* mRNA in HepG2 cells (Figure 6E). Of note, XI-011-induced PARP cleavage was decreased following siRNA-mediated MDM4 depletion in HepG2 cells compared with siNS-transfected control cells, indicating that apoptosis induction by XI-011 requires MDM4 expression (Appendix A). Although HLE cells harbor a mutant *p53* gene (p.R249S), XI-011 treatment also resulted in upregulation of *p21* mRNA and induction of PARP cleavage in these cells (Figure 6D,E). After inhibition of protein biosynthesis by cycloheximide (CHX) treatment, the half-life time of the wild-type p53 protein was increased in HepG2 cells following XI-011 treatment compared to dimethylsulfoxide (DMSO)-treated control cells (Figure 6F). In contrast, the half-life time of mutant p53 in HLE cells was not affected by XI-011 (Figure 6F), confirming that mutant p53 may escape from proteasomal degradation induced by MDM2-MDM4 heterodimers, as previously reported from other cancer entities [30]. However, *p53* mRNA levels were not affected by XI-011 treatment in HepG2 cells, whereas increased *p53* gene expression was observed in HLE cells treated with 0.2 and 0.5 µM XI-011, which contrasts with the unaltered p53 protein levels of this cell line (Appendix A), suggesting the possibility that the p53 variant R249S has not completely lost its function to transcriptionally activate (some) p53 target genes. Additionally, XI-011 treatment significantly decreased the viability of HepG2 and HLE cells compared to DMSO-treated controls in a dose- and time-dependent manner (Figure 6G), similarly to reduced cell growth observed following MDM4 depletion in vitro and in vivo [3,20]. Of note, siRNA-mediated p53 inhibition in combination with XI-011 treatment lowered the *MDM4* gene expression and consequently reduced the viability of HepG2 and HLE cells compared to the corresponding controls, excluding the possibility that the observed XI-011-driven biological effects were mediated by p53 (Appendix A). Since we observed that XI-011 reduced *MDM4* transcription, we hypothesized that XI-011 may affect the machinery driving *MDM4* transcription. Indeed, SRF, ELK1, and ELK4 protein levels were significantly reduced upon XI-011 treatment (0.5 and 1 μM) in both HCC cell lines (Figure 6H). Thus, XI-011 reduced *MDM4* expression by targeting the central TFs required to activate *MDM4* transcription. Importantly, XI-011 did not affect the protein half-life time of the TFs (Appendix A), suggesting that the observed downregulation of transcription factors was not due to increased proteasomal degradation. In line with this, the expression of a selection of canonical SRF targets (*VCL1*, *VIM*, *BCL2*) [31] was reduced following XI-011 treatment in HLE cells compared to DMSO-treated control cells (Appendix A), while *ACTB* mRNA levels were not significantly affected by the same treatment (Appendix A). Additionally, *c-MYC* expression, which was found upregulated in *SRF-VP16^iHep^* mice, was significantly diminished upon XI-011 treatment (Appendix A). Furthermore, SRF overexpression was able to rescue the MDM4 protein levels following XI-011 treatment again supporting the central role of SRF in driving *MDM4* transcription in HCC cells (Appendix A).

## 3. Discussion

MDM4 upregulation has been reported in various human cancers [18,20]. In human HCC, this can be in part explained by genomic amplification of the *MDM4* gene locus at chromosome 1q32 as well as post-translational stabilization in the context of activated AKT/mTOR signaling [3,7,20]. Here, we reported *MDM4* upregulation due to aberrant transcriptional activation in HCC. Our cell- and molecular-biological analyses demonstrated that the positive association between the mRNA levels of *MDM4* and its putative TFs, *SRF*, *ELK1*, and *ELK4*, can be explained by their concerted binding to and activation of the *MDM4* promoter in human HCC. Our findings are in line with a previous study demonstrating that activated KRAS proto-oncogene and insulin-like growth factor 1 signaling induce *MDM4* expression at least partially via ELK1 in breast, colon, and lung cancer cells [32]. Furthermore, we demonstrated that SRF is constitutively present in TF complexes driving *MDM4* expression and is supported by different ETS family proteins in human HCC cells. A previous study demonstrated the interchangeability of ETS cofactors in the formation of SRF ternary complexes [33] and we have provided evidence that ELK4 may functionally substitute ELK1 in a cell context-dependent manner, as shown by the dominant enrichment of ELK4 at the *MDM4* promoter in HepG2 cells.

The biological relevance of SRF expression for maintaining liver homeostasis was demonstrated in a liver-specific *SRF* knockout mouse, which revealed a severely impaired liver function, hepatocyte proliferation, and survival upon loss of SRF-dependent transcription [34]. SRF has been shown to bind to CArG box motifs (CC(A/T)_6_GG) in the promoter of genes that are expressed in response to mitogenic signaling [34,35]. The target gene specificity of SRF is determined by transcriptional co-factors, in particular members of the TCF family (e.g., ELK1, ELK4, and NET, all responding to activated RAS signaling) and the MRTF family (MRTF-A and MRTF-B, mediating Rho-Actin pathway activation). The modulation of MRTF family members did not affect *MDM4* mRNA levels (personal observation, 2020). In contrast, both FCS-induced phosphorylation of ERK1/2 in vitro and expression of constitutively active SRF-VP16 in vivo resulted in an upregulation of MDM4 protein levels, which promoted malignant transformation in *SRF-VP16^iHep^* mice [21]. Interestingly, the transcriptional profile of SRF-VP16-triggered murine HCC overlaps with the gene expression landscape of the molecular subclasses G1 and G2 of human HCC [36]. These subclasses are prone to *p53* mutations. However, *Ctnnb1* gene mutations occurred in about 50% of the SRF-VP16-driven HCC, a genetic event that is nearly mutually exclusive with *p53* mutation in human HCCs [7]. Thus, functional inactivation of p53 via MDM4 upregulation may functionally substitute for a *p53* gene mutation during spontaneous hepatocarcinogenesis in *SRF-VP16^iHep^* mice. Targeting SRF, ELK1, and ELK4 with specific siRNAs significantly decreased *MDM4* mRNA in human HCC cells independent of the *p53* gene status. However, it remains elusive which cellular factors determine whether ELK1 or ELK4 are preferentially selected as the co-factor for the transcriptional activation of *MDM4* and whether this has any biological or therapeutic relevance. The impact of the SRF network on *MDM4* transcription was further supported by expression analyses of *MDM4* and its *TFs* in three independent cohorts of human HCCs. Notably, the *ELK4* gene is located in close vicinity to *MDM4* at chromosomal band 1q32.1, which is the most frequently gained chromosomal region in human HCC [37]. Thus, genomically clustered oncogenes may act in a concerted network in human HCC. It remains to be determined whether MAPK signaling-mediated upregulation of *MDM4* expression has a prognostic impact on HCC patients after liver resection (Figure 1D), as we were not able to independently validate this finding using the TGCA data set (Appendix A), suggesting the possibility of a cohort-related bias. Nevertheless, subgroups of HCC patients may benefit from targeting the *MDM4* transcription, arguing for HCC patients to be included in early clinical studies evaluating the clinical potential of pharmacological MDM4 inhibitors. Cell-based high content drug screening recently revealed that XI-011, a pseudourea derivate, may reactivate p53 function by targeting *MDM4* transcription in human melanoma and breast cancer cells, thereby decreasing cancer cell viability. Importantly, XI-011 showed a very low cytotoxicity on normal cells [29,38]. Extending these data, we showed that XI-011 significantly decreased *MDM4* mRNA levels in HCC cell lines, independent of the *p53* status. Although induction of apoptosis was observed in both HepG2 (wild type) and HLE (R249S) cells, restoration of p53 transcriptional activity was only detected in XI-011-treated HepG2 cells, suggesting that MDM4 has a yet unknown p53-independent tumor suppressive function, which was also detectable in *p53*-depleted Hep3B cells. Recently, Miranda et al. demonstrated that XI-011 decreased *MDM4* expression in breast cancer cells harboring mutant *p53*, which inhibited tumor cell growth by activating p27 [39]. A similar mechanism may be relevant for the tumor-suppressive effect observed after targeting *MDM4* in *p53*-mutated HCC cells in vitro and in vivo [20]. The mechanism by which XI-011 targets *MDM4* transcription is still only partly understood, but our data from p53-depleted cells suggest that its mode of action is complex and may not be completely mediated by *MDM4* inhibition. However, our study showed, for the first time, that XI-011 reduces the expression of central TFs required for *MDM4* transcription in human HCC cells (Figure 6H), which consequently leads to reduced *MDM4* expression levels. In line with this, it has been shown that XI-011 treatment in MCF7 breast cancer cells dramatically reduces the binding of RNA Pol II to the *MDM4* promoter, and thus the rate of *MDM4* transcription [40].

## 4. Materials and Methods

### 4.1. Human Tissue Samples

Expression profiles were generated from 37 human HCCs, as described previously [4]. The specimens included 22 resection specimens and 15 explant livers. The median age at surgery was 57 years (range 16–78) and the male/female ratio was 3:1. Human tissue samples were provided by the Tissue Bank of the National Center for Tumor Diseases Heidelberg. According to the vote, informed consent was not required because only long-term archived (>5 years), pseudonymized samples were used for this study. From three patients, two HCC nodules were included that previously showed different aCGH data, indicating independent tumor development. Etiology was determined as previously described [3]. The underlying etiologies were HBV (*n* = 8), HCV (*n* = 9), alcohol (*n* = 6), cryptogenic (*n* = 11), genetic hemochromatosis (*n* = 2), and α1-antitrypsin deficiency (*n* = 1). Patient characteristics are shown in Appendix A.

Additionally, an independent cohort of 32 HCC tissues harboring wild-type p53 and corresponding surrounding non-neoplastic liver tissues (SL) were analyzed [41]. Characteristics of these patients are shown in Appendix A. The latter liver tissues were kindly provided by Snorri S. Thorgeirsson (National Cancer Institute, Bethesda, MD, USA). The samples were exempted by Institutional Review Board Approval as they have been provided in a pseudonymized form by the National Cancer Institute (NCI, Bethesda, MD, USA).

For the analysis of the TCGA HCC cohort (LIHC), gene expression and mutation data and curated clinical outcome data were downloaded from the PanCanAtlas website (https://gdc.cancer.gov/about-data/publications/pancanatlas). A total of 377 liver hepatocellular carcinoma patients with all three kinds of data available were included in the analysis. Tumors with non-synonymous *p53* mutations in the coding region or mutations at splice sites were considered as *p53* mutated, all other tumors were considered as *p53* wild type.

### 4.2. SRF Transgenic Mice

SRF-VP16 transgenic mice (*SRF-VP16^iHep^*, Mus musculus, C57131/6, males), which spontaneously develop HCC, were previously described [21]. Animal housing and handling was in accordance with the Federation of European Laboratory Animal Science Associations and approved by the local ethics committee (Project IM1/14, Regierungspräsidium Tübingen, Tübingen, Germany). Tissue samples were collected from 30- and 40-week-old mice. Nodular and tumor samples were isolated when visible in resected liver tissues. No randomization was applied and no blinding was done.

### 4.3. Cell Lines and siRNA or Plasmid Transfection

HepG2 cells (ATCC) and Hep3B (ATCC) were cultured in RPMI and MEM medium respectively, while HuH7 (ATCC) and HLE (JCRB) cell lines in DMEM medium (Thermo Fisher Scientific, Waltham, MA, USA). Culturing media were supplemented with 10% fetal bovine serum (Thermo Fisher Scientific) and 1% penicillin–streptomycin (Thermo Fisher Scientific) at 37 °C (5% CO_2_), and cells were passaged every 3–4 days. All the cell lines were tested routinely for mycoplasma contamination. STR profiling was performed for authentication of HCC cell lines. HLE cells were transiently transfected with a pCMV6-AC-GFP vector containing either a full-length human SRF cDNA (RG208596 from OriGene Technologies, Rockville, MD, USA), or a pCGN vector containing a full-length human ELK1 cDNA, an inactive ELK1 variant (ELK1 S383A cDNA), or a pCS2plus vector containing a SRF-VP16 cDNA [26] following the manufacturer’s protocol, using Lipofectamine 3000 (Invitrogen, Karlsruhe, Germany). pCGN-ELK-1 and pCGN-ELK-1 S383A were a gift from Ron Prywes Lab (Addgene plasmids #27156 and #27160) [42]. All siRNA transfections were performed using Oligofectamine (Invitrogen) according to the manufacturer’s protocol. The sequences of the siRNAs (Eurofins MWG Operon, Ebersberg, Germany; pre-designed siELK4_2 from Thermo Fisher Scientific (Waltham, MA, USA) and the final concentrations used for transfection are listed in Appendix A. For RNA or protein isolation, cells were collected 48 or 72 h after siRNA transfection. For luciferase reporter assays, HCC cells were plated in 12-well plates and transfected with siRNAs against either SRF, ELK1, or ELK4. Then, 24 h after siRNA transfection, cells were transfected with 1 µg of an *MDM4* promoter reporter plasmid (HPRM23227, Gene Copoeia, Rockville, MD, USA), using Lipofectamine 3000 (Invitrogen) according to the manufacturer’s recommendations. Medium was collected after 48 h and the luciferase promoter activity was measured using the Secrete-Pair™ Dual Luminescence Assay Kit (Gene Copoeia) in a FLUOstar Omega Microplate Reader (BMG Labtech, Ortenberg, Germany) before cells were scratched for protein or RNA isolation followed by Western blotting analysis or qPCR, respectively.

### 4.4. XI-011 and Pharmacological Pathway Inhibitor Treatment of HCC Cell Lines

HCC cells were plated at a density of 2 × 10^5^ cells/well in 6-well plates and were incubated with XI-011 ([10-methyl-9-anthryl]methyl imidothiocarbamate or NSC146109; Tocris Wiesbaden-Nordenstadt, Germany) at different time points as indicated in the figures. To evaluate protein half-life time, cells were treated with 1 µM XI-011 for 16 h before adding cycloheximide (450 µM; Santa Cruz Biotechnology, Santa Cruz, CA, USA). For pharmacological inhibition of the PI3K or ERK pathways, respectively, cells were seeded as reported above and starved overnight with FCS-deprived medium. The following day, cells were stimulated by adding FCS in combination with DMSO, LY294002 for 4 h (50 µM, PI3K inhibitor, Ann Arbor, MI, USA) or LY3214996 for 8 h (5 µM, ERK1/2 inhibitor, Selleckchem, Munich, Germany), respectively. Then, cells were collected and RNA and protein were isolated. T0 control cells were collected after overnight starvation before FCS and drugs were added. Cell viability was determined using a standard MTT assay (Methylthiazolyldiphenyl-tetrazolium bromide, Sigma). Briefly, HepG2 and HLE cells were plated at a density of 3 × 10^4^ cells/well in 96-well plates and treated with XI-011 (Tocris) at different concentrations and time points, as indicated in the graphs. After adding DMSO/EtOH solution (1:2) to each well, the colorimetric detection was carried out in a FLUOstar Omega Microplate Reader (BMG Labtech).

### 4.5. Western Immunoblotting

Cells were homogenized in lysis buffer (Cell Signaling, Frankfurt, Germany) supplemented with Protease Inhibitor Mix M (Serva, Heidelberg, Germany) and PhosStop Phosphatase Inhibitor Cocktail (Merck, Darmstadt, Germany) and were sonicated subsequently. Protein concentrations were determined by NanoDrop. For Western immunoblotting, 140 μg aliquots of cell lysate were denatured by boiling in SDS sample buffer, separated by SDS-PAGE, and blotted onto 0.45 µm (GE Life Sciences, Freiburg, Germany) nitrocellulose membranes, respectively. Membranes were blocked with either 5% nonfat dry milk in Tris-buffered saline or Odyssey TBS Blocking Buffer (Li-Cor, Li-Cor Biosciences, Lincoln, NE, USA) containing 0.1% Tween 20 for 1 h and probed overnight with specific antibodies, as listed in Appendix A. Each primary antibody incubation was followed by treatment with an IRDye-labelled secondary antibody for 1 h (1:10,000; Li-Cor) and visualized using the Odyssey Imager or Odyssey CLx (Li-Cor). Densitometric analyses were carried out using Image Studio software (Li-Cor); more in detail, each specific protein signal was first normalized against the corresponding loading control (ACTIN, GAPDH or VINCULIN), then compared to the corresponding normalized control sample for data included in the histograms.

### 4.6. Immunohistochemistry Analysis

Murine liver tissues were formalin-fixed and embedded in paraffin using standard techniques; immunohistochemistry was performed on 3-µm sections. Hematoxylin eosin staining was performed as previously described [20]. Antigen retrieval was carried out using antigen retrieval solution pH 9 for MDM4 and ELK4 or pH 6.1 for pELK1, respectively (Dako, Glostrup, Denmark). Antibodies are listed in Appendix A. Visualization was done using the EnVision method (Dako) and counterstaining was performed using hemalum solution.

### 4.7. RNA Isolation, cDNA Synthesis, and Quantitative Real-Time Reverse-Transcription Polymerase Chain Reaction

RNA from human and murine samples was isolated using 100 mg of snap-frozen tissue with the RNeasy Mini-Kit (QIAGEN, Hilden, Germany). RNA for cell-line experiments was obtained using the NucleoSpin RNA kit (Macherey-Nagel, Düren, Germany) following the manufacturer’s instructions. cDNA synthesis and quantitative real-time reverse-transcription polymerase chain reaction was performed as reported previously [43]. Briefly, 1 µg of total RNA was converted to cDNA by using RevertAid H Minus reverse transcriptase (Thermo Scientific); 4 ng of cDNA was mixed with the specific primer pairs (Microsynth AG, Balgach, Switzerland) and 1x PowerUp SYBR Green Master Mix (Thermo Fisher Scientific) in a 96-well plate. *GAPDH* or *18S rRNA* was used as endogenous control. Quantitative real-time PCRs were performed in a StepOne Plus device (Thermo Fisher Scientific). Primer sequences are listed in Appendix A.

### 4.8. DNA Microarray Hybridization and Analysis

Quality and integrity of the total RNA was controlled using an Agilent Technologies 2100 Bioanalyzer (Agilent Technologies, Waldbronn, Germany). Total RNA (200 ng) was applied for Cy3-labelling reaction using the one color Quick Amp labelling protocol (Agilent Technologies). Labelled cRNA was hybridized to Agilent human 8 × 60 k microarrays at 68 °C for 16 h and scanned using the Agilent DNA Microarray Scanner (Agilent Technologies). Expression values were calculated by the software package Feature Extraction 10.5.1.1. Complete data are available online (http://www.ncbi.nlm.nih.gov/geo/query/acc.cgi?acc=GSE50579). Centralized gene expression levels were calculated after normalization of the raw expression data of each HCC against the mean expression of the gene of interest in seven normal liver samples (NLs).

### 4.9. Chromatin Immunoprecipitation (ChIP)

For chromatin immunoprecipitations, HLE, HuH7, or HepG2 cells were seeded in 15 cm dishes and collected when confluent. Chromatin isolation and chromatin immunoprecipitation were performed using the SimpleChIP Enzymatic Chromatin IP Kit (Cell Signaling) following the manufacturer’s protocol. For each sample, 2 µg of a specific antibody against SRF, ELK1, or ELK4 was used, as listed in Appendix A After purification, DNA obtained was quantified by quantitative real-time reverse-transcription polymerase chain reaction using a StepOne Plus device (Thermo Fisher Scientific). Equal amount of input sample was used in all the experiments and ChIP-qPCR Ct data for target (positive) and nontarget (negative, IgG) sequences were normalized against input (ΔCt). Next, fold enrichment of the specific TF sequence in ChIP DNA (ΔCt positive) over the negative locus (ΔCt negative) was calculated (2ΔΔCt, where ΔΔCt = ΔCt positive − ΔCt negative). Primer sequences are listed in Appendix A.

### 4.10. Statistical Analysis

Statistical analyses were performed using Graph Pad Prism 8.02 (GraphPad Software, San Diego, CA, USA) or Excel (Excel 2010, Microsoft, Redmond, WA, USA). Statistical significance was evaluated by using two-tailed Mann–Whitney U or Wilcoxon test. When more than two experimental groups were compared, either Kruskal–Wallis test with Dunn´s test or, in case of a normal distribution defined by Kolmogorov–Smirnov or Shapiro–Wilk test, one-way or two-way ANOVA followed by Tukey´s test were used. Expression data and all data derived from biological replicates are represented as mean with SEM. The association between *MDM4* and *SRF*, *ELK1*, *ELK4* expression in HCC samples was measured by Spearman’s rank correlations. Overall survival, defined as the time interval between diagnosis and death, was used as clinical endpoint for survival analyses as recommended by the TCGA clinical data resource [44]. The median expression of each investigated biomarkers was used as cut-off to stratify HCC patients. Univariate survival analysis was based on the Kaplan–Meier method and a Cox regression model was used for the survival curve analyses. A *p*-value < 0.05 was considered statistically significant.

## 5. Conclusions

In conclusion, our study demonstrated that transcriptional dysregulation promotes the oncogenic function of MDM4 in human HCC, which is associated with a shorter survival probability of HCC patients, and suggests that targeting *MDM4* transcriptionally may provide a rationale for precision therapy of human HCC. Importantly, such an approach would be expected to be effective, independent of both the copy number of the *MDM4* gene and the mutational status of the *p53* gene.

## Figures and Tables

**Figure 1 cancers-13-00199-f001:**
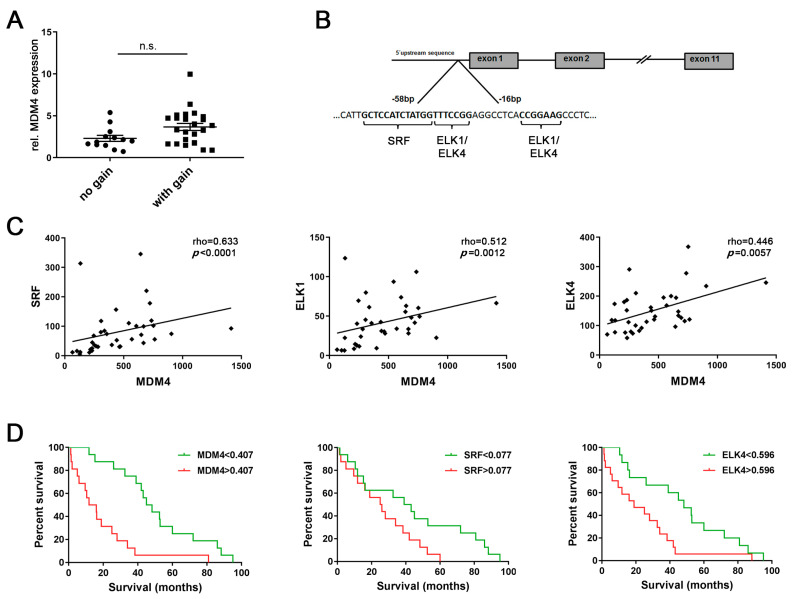
Aberrant transcriptional activation may be involved in upregulation of mouse double minute homolog 4 (*MDM4*) in human HCC. (**A**) Relative *MDM4* mRNA expression in human hepatocellular carcinomas (HCCs) with balanced (*n* = 13) and gained *MDM4* (*n* = 24) gene loci, respectively. Mann–Whitney U test: *p* > 0.05. (**B**) An in silico analysis of the basal *MDM4* promoter region identified putative transcription factor binding sites for serum response factor (SRF), ELK1, and ELK4. (**C**) Expression profiling revealed a positive association between *MDM4* mRNA and the expression level of the putative transcription factors *SRF*, *ELK1*, and *ELK4* in human HCC samples (*n* = 37). (**D**) *MDM4*, *SRF*, and *ELK4* mRNA levels were associated with the survival probability of HCC patients in a second cohort (*n* = 32). Each median expression level was used for stratification. Abbreviation: n.s., not statistically significant.

**Figure 2 cancers-13-00199-f002:**
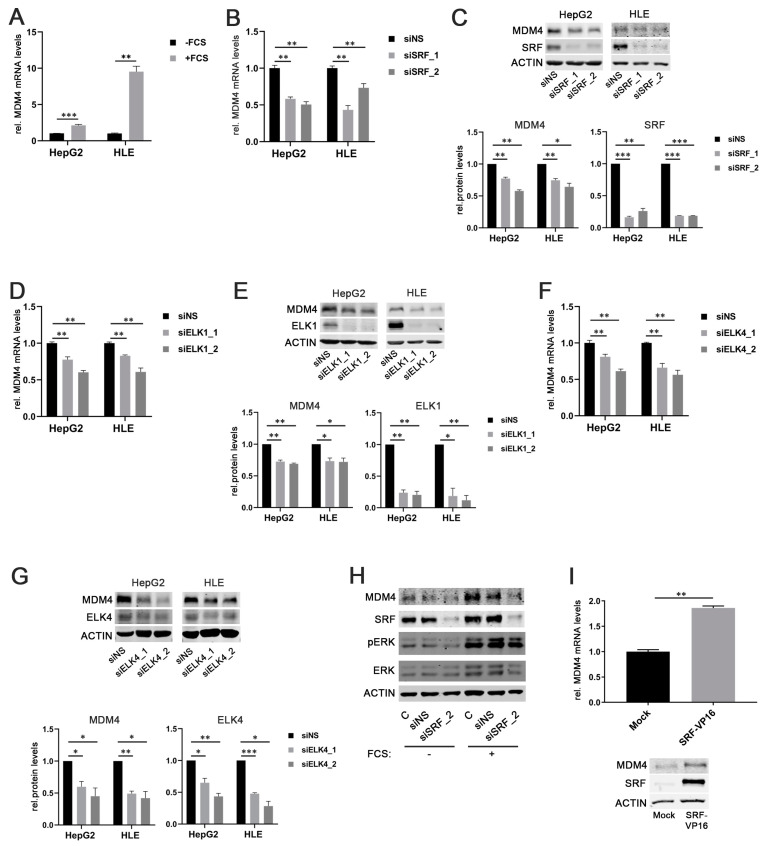
SRF, ELK1, and ELK4 regulate MDM4 expression in HCC cell lines. (**A**) Increased *MDM4* mRNA levels in HepG2 and HLE cell lines following fetal calf serum (FCS) stimulation compared to starved control cells. (**B**) MDM4 mRNA and (**C**) protein levels following siRNA-mediated knockdown of SRF compared to control cells transfected with a scrambled, nonsense siRNA (siNS) in HepG2 and HLE cells. (**D**) MDM4 mRNA and (**E**) protein levels following siRNA-mediated knockdown of ELK1 compared to control cells transfected with a scrambled, nonsense siRNA (siNS) in HepG2 and HLE cells. (**F**) MDM4 mRNA and (**G**) protein levels following siRNA-mediated knockdown of ELK4 compared to control cells transfected with a scrambled, nonsense siRNA (siNS) in HepG2 and HLE cells. (**H**) siRNA-mediated knockdown of SRF (siSRF_2) prevents FCS-stimulated MDM4 protein upregulation. (**I**) MDM4 mRNA and protein expression 48 h following transfection of HuH7 cells with an SRF-VP16 expression vector compared to mock transfected control cells. Original western blots are shown in Appendix A. Data are presented as mean ± SEM. Mann–Whitney U test: * *p* < 0.05, ** *p* < 0.01, *** *p* < 0.001. Abbreviations: siNS—scrambled, nonsense siRNA; siSRF_1/_2, siELK1_1/_2, siELK4_1/_2—siRNA 1 and 2 specifically targeting SRF, ELK1, and ELK4, respectively.

**Figure 3 cancers-13-00199-f003:**
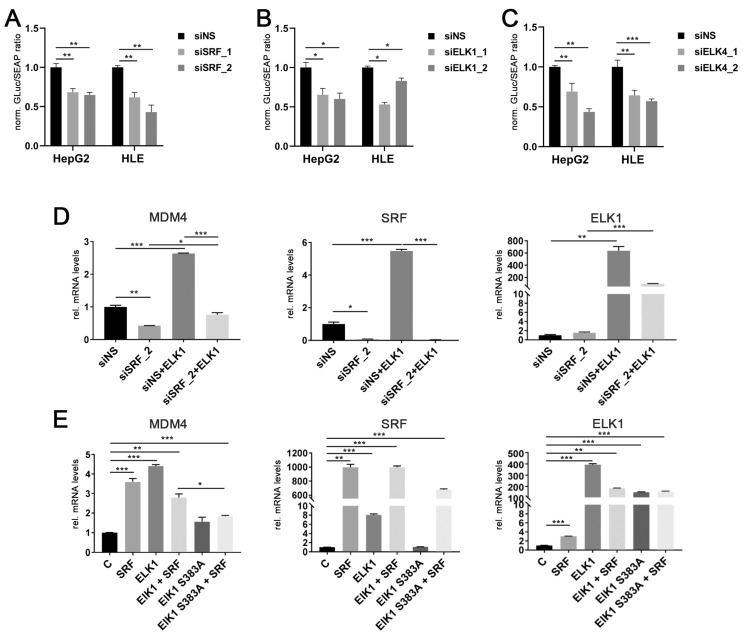
ELK1 and ELK4 are essential co-factors for SRF-mediated transcriptional regulation of *MDM4* in HCC. Luciferase activity of a *MDM4* promoter reporter upon siRNA-mediated knockdown of (**A**) SRF, (**B**) ELK1, and (**C**) ELK4 in HepG2 and HLE cells compared to controls. (**D**) *MDM4* mRNA levels after co-transfection of an *ELK1* cDNA with siNS or siSRF. Transfection efficacy was confirmed by detection of *ELK1* and *SRF* mRNA levels. (**E**) *MDM4* mRNA levels following transfection of the indicated cDNA plasmids. ELK1 S383A represents an inactive variant, which cannot be activated by phosphorylation of S383 and is thus unable to initiate target gene transcription. Transfection efficacy was confirmed by detection of *ELK1* and *SRF* mRNA levels. Data are presented as mean ± SEM. Mann-Whitney U test: * *p* < 0.05, ** *p* < 0.01, *** *p* < 0.001. Abbreviations: siNS, scrambled, nonsense; siSRF_1/_2, siELK1_1/_2 siELK4_1/_2, siRNA 1 and 2 specifically targeting SRF, ELK1 and ELK4, respectively; ELK1, ELK1 cDNA; ELK1 S383A, ELK1 S383A cDNA; GLuc, *Gaussia* luciferase; SEAP, Secreted Alkaline Phosphatase; norm., normalized against control.

**Figure 4 cancers-13-00199-f004:**
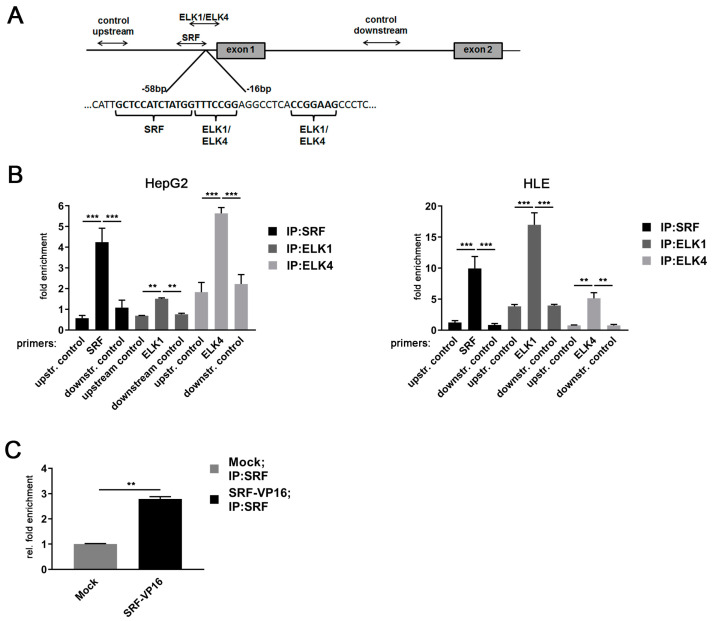
*MDM4* gene promoter is activated by an SRF-ETS family transcription factor complex in HCC cell lines. (**A**) Schematic representation of the positioning of primers used for ChIP analyses of the *MDM4* promoter region. (**B**) Specific binding of SRF, ELK1, and ELK4 at their cognate binding sites in the *MDM4* promoter compared to control primers located either down- or upstream of the predicted basal *MDM4* promoter as detected by quantitative real-time PCR of immunoprecipitated chromatin. (**C**) Relative enrichment of SRF-immunoprecipitated DNA in HuH7 cells transfected with SRF-VP16 expression plasmid compared to mock transfected control cells. Data are presented as mean ± SEM. Mann–Whitney U test: ** *p* < 0.01, *** *p* < 0.001. Abbreviations: upstr. control, control primer amplifying a region upstream of the *MDM4* promoter; downstr. control, control primer amplifying a region downstream of the *MDM4* promoter.

**Figure 5 cancers-13-00199-f005:**
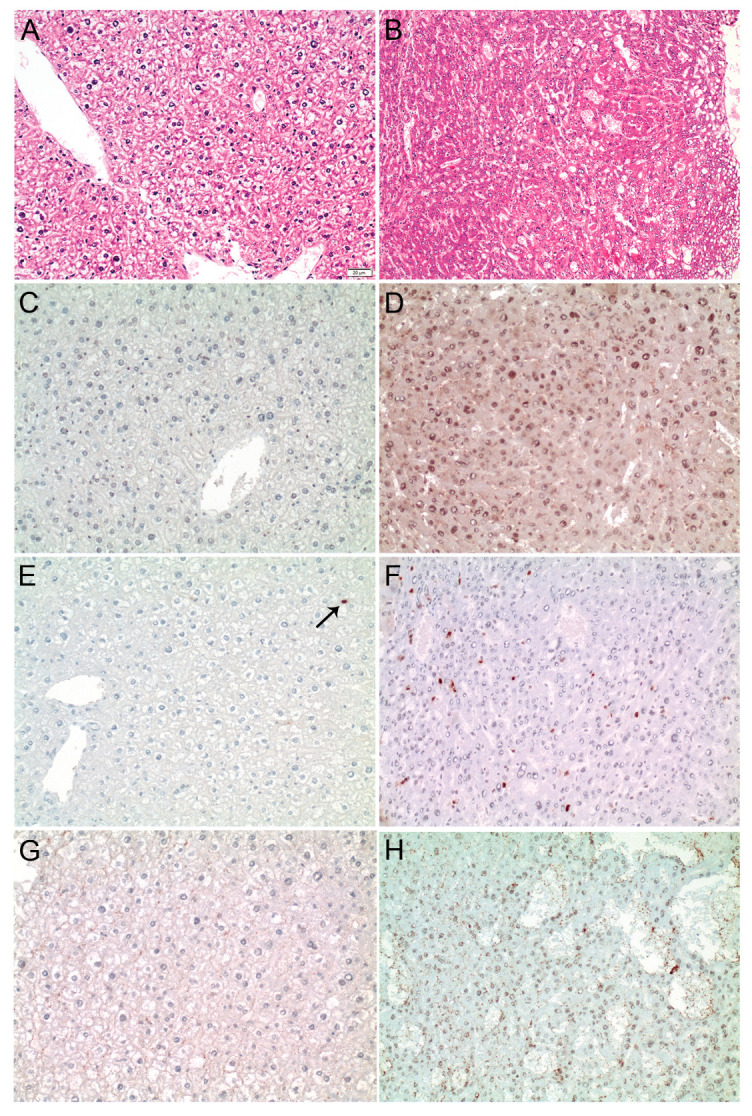
MDM4 is upregulated in *SRF-VP16^iHep^* transgenic mice. (**A**) Normal liver parenchyma in control mice (HE staining). (**B**) Well-differentiated HCC in a 30-week-old *SRF-VP16^iHep^* mouse showing trabecular disarray and pseudogland formation. MDM4 immunostaining is negative in control mice (**C**), while a diffuse, predominantly nuclear staining is seen in *SRF-VP16^iHep^* mice (**D**). Individual hepatocyte nuclei (arrow) are positive for phosphorylated-ELK1 in the control liver (**E**), whereas the number of p-ELK1 positive nuclei is significantly increased in *SRF-VP16^iHep^* mice (**F**). There is no ELK4 immunosignal in control mice (**G**). In contrast, the SRF-VP16-induced HCC reveals weak to moderate nuclear ELK4 staining (**H**). Scale bar: 20 µM.

**Figure 6 cancers-13-00199-f006:**
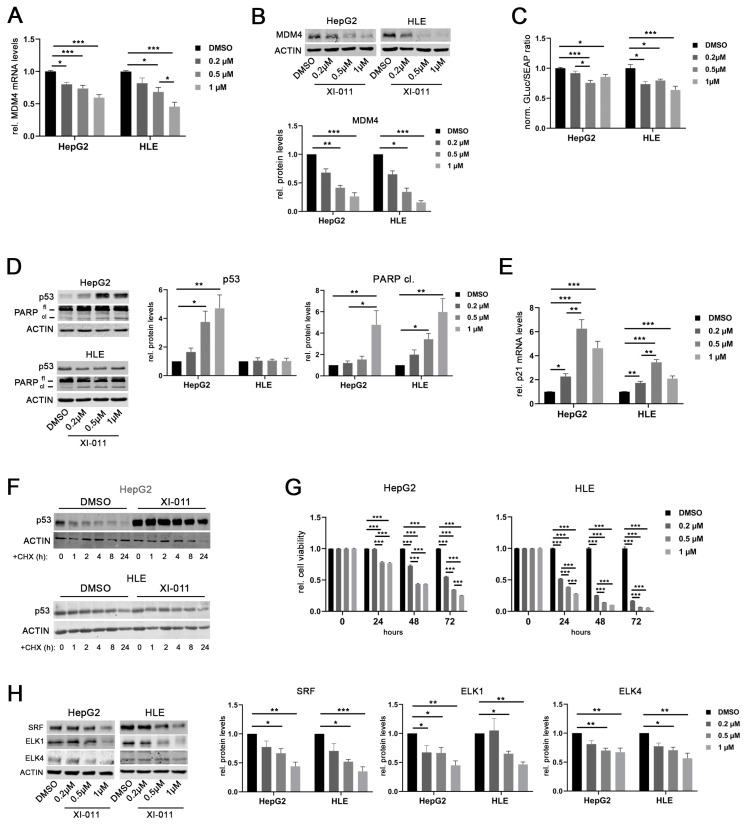
XI-011 inhibits *MDM4* transcription by transcription factor downregulation. (**A**) *MDM4* mRNA level at 16 h following XI-011 treatment of HepG2 and HLE cells. (**B**) MDM4 protein expression as detected by Western immunoblots and corresponding densitometric analysis (lower panel) 16 h following XI-011 treatment of HepG2 and HLE cell lines. (**C**) Luciferase activity of a *MDM4* promoter reporter following XI-011 treatment in HepG2 and HLE cells, respectively. (**D**) Induction of p53 and PARP protein cleavage following XI-011 treatment of HCC cell lines as indicated. (**E**) XI-011 treatment led to *p21* mRNA induction in both HCC cell lines. (**F**) p53 protein expression over time following XI-011 treatment (1 µM) with or without additional cycloheximide (CHX) treatment in *p53*-wildtype HepG2 and *p53*-mutant HLE cells as detected by Western immunoblotting. (**G**) Relative cell viability of HepG2 and HLE cells over time after XI-011 treatment using the indicated doses compared to control cells. (**H**) Western immunoblots following XI-011 treatment of HepG2 and HLE cells. Original western blots are shown in Appendix A. Data are presented as mean ± SEM. One-way and two-way ANOVA with Tukey’s test were used in panel C and G, respectively; all the other data were analyzed by Kruskal–Wallis followed by Dunn’s test: * *p* < 0.05, ** *p* < 0.01, *** *p* < 0.001. Abbreviations: fl, full-length PARP protein; cl, cleaved PARP protein; GLuc, *Gaussia* luciferase; SEAP, secreted alkaline phosphatase; norm., normalized against control; CHX, cycloheximide.

## Data Availability

Publicly available datasets were analyzed in this study. These data can be found here: http://www.ncbi.nlm.nih.gov/geo/query/acc.cgi?acc=GSE50579 and https://gdc.cancer.gov/about-data/publications/pancanatlas.

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
