# Peer review of "Serum Response Factor (SRF) Drives the Transcriptional Upregulation of the MDM4 Oncogene in HCC"

_cancers, 2021, doi:10.3390/cancers13020199_

Round 1
Reviewer 1 Report
The study carried out by Pellegrino et al. investigates the transcriptional regulation of MDM4, its relationship with p53 expression in hepatocellular carcinoma (HCC), and evaluates the utility of MDM4 as a pharmacological target against HCC. The work is well written and easy to understand and follow. The methodology is well explained, and the experimental designs are correct. The conclusion is mostly supported by the results.
I have some considerations to improve the article:
1) Inactivation of p53 frequently occurs in hepatocellular carcinoma due to inactivating mutations or deletions in the TP53 gene. In these cases, MDM4 would not be useful as a pharmacological target to reactivate p53. In survival studies with HCC patients according to the MDM4, SRF and ELK4 expression would be interesting to separate patients in those who have lost p53 and those who have not.
2) In this sense, a group of patients who have gain-of-function mutations can also be separated in survival studies according to the expression of MDM4.
3) As shown in Figure 6E, the inhibitor XI-011 was apparently not able to increase p53 levels in a concentration-dependent manner in the HLE cells. The authors argue that this could be due to the stability of the protein. Furthermore, the p53-mediated effects on HEL cells seem to be as expected. The authors could measure the p53 mRNA levels in HEL cells to demonstrate the expected increase mediated by the XI-011 inhibitor.
4) Is there any evidence that the tumors have been generated by an alteration of the p53 pathway, e.g. downregulation of p53 or p21?
Author Response
Point-by-point reply to the reviewer´s questions
Reviewer 1
The study carried out by Pellegrino et al. investigates the transcriptional regulation of MDM4, its relationship with p53 expression in hepatocellular carcinoma (HCC), and evaluates the utility of MDM4 as a pharmacological target against HCC. The work is well written and easy to understand and follow. The methodology is well explained, and the experimental designs are correct. The conclusion is mostly supported by the results.
I have some considerations to improve the article:
1) Inactivation of p53 frequently occurs in hepatocellular carcinoma due to inactivating mutations or deletions in the TP53 gene. In these cases, MDM4 would not be useful as a pharmacological target to reactivate p53. In survival studies with HCC patients according to the MDM4, SRF and ELK4 expression would be interesting to separate patients in those who have lost p53 and those who have not.
We thank the reviewer for the valuable comments. The frequency of p53 mutations and the mutation spectra in human hepatocellular carcinoma (HCC) vary substantially between different geographic regions and presumably reflect relevant differences in etiology and host oncogenic mechanisms. Whereas the frequency of p53 mutations in HCCs is high in sub-Saharan Africa (>50%) and Southeast Asia (>35%) (www-p53.iarc.fr/), it seems to be less than 10% in Western countries. We have previously reported that overexpression of MDM4 may lead to functional inactivation of p53 and may thus substitute for p53 mutations (Schlaeger et al., Hepatology 2008). In addition, we have previously shown both in vitro and in vivo that MDM4 exerts oncogenic functions in HCC independently from the p53 gene status (Schlaeger et al., Hepatology 2008; Pellegrino et al., Hepatology 2014). In line with this, we demonstrate a reduced cell viability (Fig. 6G) and increased apoptosis rate (Fig. 6D) of both p53-wild-type HepG2 and -mutated (R249S) HLE cells upon XI-011-mediated targeting of MDM4. In addition, XI-011 treatment reduced MDM4 expression in p53-deficient Hep3B cells leading to apoptosis (line 266 and Fig. S5B/C). Together, the data suggest that targeting MDM4 could be beneficiary to HCC patients independent of the p53 gene status.
Unfortunately, we do not have survival data available for HCC cohort shown in Fig. 1A/C and all patients shown in Fig. 1D suffered from HCC with a p53 wild-type gene. In order to address the point raised by the reviewer, we additionally analyzed the TCGA data set (LIHC cohort). As shown in Fig. S2E, the stratification of HCC patients in wild-type and mutated cases of this cohort is hampered by the fact that p53-mutated cases only show a trend for reduced overall survival (HR 1.45 (0.99-2.11), p=0.054). Thus, further stratification in p53 wild-type respectively mutated cases with low and high expression of MDM4, SRF, and ELK4 also failed to provide significant differences in terms of survival in the four groups (Fig. S2F). We have added the issue of a potential cohort-related bias to the discussion (line 363).
2) In this sense, a group of patients who have gain-of-function mutations can also be separated in survival studies according to the expression of MDM4.
Gain-of-function p53 mutations do not represent a homogeneous group. Basically, all p53 missense mutations promote a diverse spectrum of loss-of-function over the remaining wild-type allele (see answer to point 3), but the majority of them promote additional gains-of- functions, which are selected during tumor development and progression and thus promote an evolutionary privilege. Unfortunately, there is no universal consensus on certain groups of gain-of-function mutations, which would allow further subgrouping of missense mutations (Soussi, 25 Years of p53 Research, Springer 2005; Brosh et al, Nature Reviews Cancer 2009; Muller et al., Cancer Cell 2014). Thus, we feel that we cannot reliably address this issue.
3) As shown in Figure 6E, the inhibitor XI-011 was apparently not able to increase p53 levels in a concentration-dependent manner in the HLE cells. The authors argue that this could be due to the stability of the protein. Furthermore, the p53-mediated effects on HLE cells seem to be as expected. The authors could measure the p53 mRNA levels in HEL cells to demonstrate the expected increase mediated by the XI-011 inhibitor.
We thank the reviewer for this interesting comment. While HepG2 cells harbor wild-type p53, this gene is mutated in HLE cells (c.747G>C, R249S). XI-011 treatment did not increase the p53 mRNA levels in HepG2 cells compared to DMSO-treated control cells (Suppl. Fig. 5I), indicating that the upregulation of p53 protein in HepG2 cells following XI-011 can be attributed to an increased stability of the p53 wild-type protein as demonstrated by CHX treatment (Fig. 6F). As shown in DMSO-treated control cells the p53 variant R249S (HLE) is more stable than the p53 wild-type protein (HepG2) and XI-011 treatment obviously does not alter the p53 protein level in HLE cells (Fig. 6F). Surprisingly, lower dose (0.2 and 0.5 µM) XI-011 treatment resulted in upregulation of p53 mRNA in HLE compared to DMSO-treated control cells (Suppl. Fig. 5I), which was paralleled by upregulation of p21 mRNA (albeit at lower level compared to HepG2 cells; Fig. 6E) suggesting the possibility that the p53 variant R249S has not completely lost its function to transcriptionally activate (some) p53 target genes and XI-011 treatment may thus be able to induce p53-dependent transcription in HLE cells to some extent as well. We have discussed this issue in the revised manuscript (line 282). As already pointed out in our answer to point 1, XI-011 treatment reduced cell viability and increased apoptosis of HCC cells independent of the p53 gene status and may thus represent an interesting pharmacological target independent of the p53 gene status for this disease.
4) Is there any evidence that the tumors have been generated by an alteration of the p53 pathway, e.g. downregulation of p53 or p21?
We thank the reviewer for this important question. As published previously, genome-wide expression profiling of murine SRF-VP16-triggered HCC showed a recurrent upregulation of genes involved in oncofetal development, cell proliferation, immune regulation, and lipid metabolism (Ohrnberger et al., Hepatology 2015). Comparison of SRF-VP16-triggered murine and human HCCs, which included the 37 samples used in this study, revealed that the transcriptional profiles of SRF-VP16-induced murine HCC primarily correspond to the subclasses G1 and G2 of human HCC. These human HCC subgroups are characterized by the expression of oncofetal genes, mitotic cell cycle dysregulation, and p53 gene mutations (Boyault et al, Hepatology 2007; Ohrnberger et al., Hepatology 2015; Calderaro et al., J Hepatol 2017). Thus, there is some evidence that the p53 signaling pathway is altered in SRF-VP16-induced murine HCC. However, SRF-VP16-induced murine HCCs are prone to CTNNB1 mutations (present in about 50% of cases), which almost exclusively occur in human HCC harboring wild-type p53 (Cancer Genome Atlas Research Network, Cell 2017). Based on these observations, we hypothesize that the increased MDM4 level functionally inhibits p53 signaling in SRF-VP16-triggered HCC. We have added this aspect to the discussion of our manuscript (line 349). In line with our previous findings, the relevance of MDM4 upregulation in human HCC was independently confirmed by the TCGA consortium demonstrating that high MDM4 expression is associated with a low p53 gene signature (Cancer Genome Atlas Research Network, Cell 2017; Pellegrino et al., Hepatology 2014).

Reviewer 2 Report
Pellegrino et al. aims to show that serum response factor (SRF) regulates the expression of the p53 repressor MDM4 in hepatocellular carcinoma (HCC). They note that genomic copy number of MDM4 does not strongly correlate with expression in HCC and therefore seek to identify regulatory elements that might contribute to MDM4 expression. An informatic analysis of the MDM4 promoter suggests that ETS family factors and SRF may contribute to MDM4 regulation in HCC. Using knockdowns, ChIP, and overexpression that authors make a convincing argument that SRF/ETS factors regulate MDM4 expression and contribute to p53 regulation in HCC. The main gap in the paper is lack of validation of the mechanism of the small molecule XI-011, which they show has strong effects on HCC cells but do not convincingly show the specificity of these effects. Should the authors address these and other more secondary concerns detailed below I believe the paper is very suitable for publication.
Major Comments
- In figure one the source and analysis of the gene expression data (figure C) is unclear. It looks like microarray data? Please expand on the analysis more by briefly describing it in the text and more extensively in the methods. Critically it is important to understand how it was normalized and if it covaries with other known prognostic factors. Specifically, I am concerned that MDM4 levels may be related to the tumor faction in the sample, which would result in positive correlations between all tumor associated transcripts.
- The SRF knockdown experiments are strong. However, I know that SRF (and ETS) knockdowns are frequently toxic to the cells. It would therefore support the authors point if they would use small molecule inhibitors of ERK, EGFR, or PI3K to block SRF activation after FCS addition to show that transient inhibition of growth factor signaling also blocks MDM4 expression. Transient expression of a dominant negative mutant SRF might also be a complimentary way to test the author’s model.
- The data on XI-011 is problematic in an otherwise strong manuscript. This is a small molecule with unknown mechanism of action, though the results are broadly consistent with the authors hypothesis their results need to be validated with known functional MDM4 inhibitors. (a) the authors should show that MDM4siRNA phenocopies XI-011 treatment, (b) show that p53 knockdown eliminates the toxicity of XI-011, (c) determine if SRF-VP16 expression can rescue the effect of XI-011, (d) determine if other SRF target expression is reduced by treatment (non-MDM4 SRF targets).
Minor Comments
Line 78 is confusing, it is not clear what the statistics refer to (mRNA vs genomic copy number)
Figure 4: What is the ChIPsignal normalized to? Are all samples normalized to the same input?
In the introduction it would be worth noting that SRF-VP16 mice develop HCC.
Typos
Line 76 – iMDM4 should be “in MDM4”
Author Response
Point-by-point reply to the reviewer´s questions
Reviewer 2
Pellegrino et al. aims to show that serum response factor (SRF) regulates the expression of the p53 repressor MDM4 in hepatocellular carcinoma (HCC). They note that genomic copy number of MDM4 does not strongly correlate with expression in HCC and therefore seek to identify regulatory elements that might contribute to MDM4 expression. An informatic analysis of the MDM4 promoter suggests that ETS family factors and SRF may contribute to MDM4 regulation in HCC. Using knockdowns, ChIP, and overexpression that authors make a convincing argument that SRF/ETS factors regulate MDM4 expression and contribute to p53 regulation in HCC. The main gap in the paper is lack of validation of the mechanism of the small molecule XI-011, which they show has strong effects on HCC cells but do not convincingly show the specificity of these effects. Should the authors address these and other more secondary concerns detailed below I believe the paper is very suitable for publication.
We thank the reviewer for the overall positive evaluation of our work.
Major Comments
1.) In figure one the source and analysis of the gene expression data (figure C) is unclear. It looks like microarray data? Please expand on the analysis more by briefly describing it in the text and more extensively in the methods. Critically it is important to understand how it was normalized and if it covaries with other known prognostic factors. Specifically, I am concerned that MDM4 levels may be related to the tumor faction in the sample, which would result in positive correlations between all tumor associated transcripts.
We apologize for providing unclear information, have revised the manuscript, and added the missing information (line 101). As assumed by the reviewer, the data shown in Figure 1C were derived from microarray profiling of human HCCs (Table 1), which have been published previously (Schlaeger et al., Hepatology 2008; Neumann et al., Hepatology 2012). As detailed in paragraph 4.8 of the Materials and Methods section, each gene expression value of every HCC sample was normalized against the mean expression of the gene of interest in the normal livers (NLs) included in the analysis. Unfortunately, we do not have survival data available for this cohort of human HCC samples and are thus not able to assess whether the MDM4 expression level represents an independent prognostic factor. However, we can rule out that the tumor fraction represents a relevant confounder as histological re-evaluation of cryo-sections was performed to ensure that only suitable samples (>90% viable and non-sclerotic tumor) were used for nucleic acid extraction (Schlaeger et al., Hepatology 2008, Neumann et al., Hepatology 2012). In addition, the prognostic relevance of high MDM4 expression was independently confirmed in several cohorts (Fig. 1D; Pellegrino et al., Hepatology 2014).
2.) The SRF knockdown experiments are strong. However, I know that SRF (and ETS) knockdowns are frequently toxic to the cells. It would therefore support the authors point if they would use small molecule inhibitors of ERK, EGFR, or PI3K to block SRF activation after FCS addition to show that transient inhibition of growth factor signaling also blocks MDM4 expression. Transient expression of a dominant negative mutant SRF might also be a complimentary way to test the author’s model.
As requested by the reviewer, we have performed additional experiments. As shown in Supplementary Fig. 3, the PI3K inhibitor LY294002 completely rescued FCS induced upregulation of MDM4 mRNA (Suppl. Fig. 3B) and protein (Suppl. Fig. 3C) in serum-starved HepG2 cells. Similar results were obtained using LY3214996 to inhibit ERK signaling (Fig. S3D/E). Thus, transient inhibition of growth factor signaling pathways blocks upregulation of MDM4 mRNA confirming our data and supporting our model.
3.) The data on XI-011 is problematic in an otherwise strong manuscript. This is a small molecule with unknown mechanism of action, though the results are broadly consistent with the authors hypothesis their results need to be validated with known functional MDM4 inhibitors. (a) the authors should show that MDM4 siRNA phenocopies XI-011 treatment, (b) show that p53 knockdown eliminates the toxicity of XI-011, (c) determine if SRF-VP16 expression can rescue the effect of XI-011, (d) determine if other SRF target expression is reduced by treatment (non-MDM4 SRF targets).
We agree that the mode of action of XI-011 is not fully characterized. We selected this compound after an extensive literature search as being the most promising compound available for targeting MDM4 transcription in vitro. Here we provide new data further characterizing this drug in our model system.
(a). Using siRNA and shRNA-mediated knockdown of MDM4 we have previously shown that targeting MDM4 has therapeutic efficacy in HCC cells both in vitro and in vivo (Schlaeger et al., Hepatology 2008, Pellegrino et al., Hepatology 2014). In particular, MDM4 inhibition led to decreased cell viability in HCC cell lines in a p53-dependent and -independent manner, which is in line with the results observed in HCC cell lines after XI-011 treatment. Since we have already provided clear evidence using both gene-specific siRNAs and shRNAs in our previous studies, we did not repeat these experiments for the current study, but we have revised the manuscript to better address this issue (line 288).
(b). From the literature it was already known that XI-001 is tumor cell-specific and does not induce genotoxic effects in isogenic normal cells (Berkson et al., Int J Cancer 2010). We have shown that XI-011 significantly decreased cell growth of HCC cells in a time- and dose-dependent manner, at least in part via the induction of apoptosis (Fig. 6G). In order to address the reviewer´s question, we have treated p53-deficient Hep3B cells (carrying a bi-allelic p53 gene deletion) with different concentrations of XI-011. As for the other HCC cells analyzed, we observed a dose-dependent inhibition of MDM4 expression, which was paralleled by PARP cleavage, indicating the induction of apoptosis (Suppl. Fig. 5 B/C). Moreover, we knocked-down p53 in HepG2 and HLE cells using gene-specific siRNAs; again, XI-011 treatment resulted in significantly reduced MDM4 mRNA and protein levels (Suppl. Fig. 5F/G). Similarly, cell viability was significantly diminished in both cell lines (Suppl. Fig. 5H). Together, the new data validate the hypothesis that targeting MDM4 promotes tumor-suppressive functions independent of the expression level and the genomic status of the p53 gene.
(c). As shown in Suppl. Fig. 6C, the XI-011-mediated knockdown of MDM4 protein expression can be rescued by overexpression of SRF in HepG2 cells in vitro, thereby demonstrating that SRF is indeed able to rescue MDM4 expression.
(d). As requested, we have determined the expression of some canonical SRF target genes. As depicted in Fig. S6B, VCL1, BCL2 and VIM were downregulated upon XI-011 treatment of HLE cells for 16 h, while ACTB expression was not affected in the same experimental setup. Additionally, the expression of the c-MYC gene was reduced as previously reported in SRF-VP16 transgenic mice (Ohrnberger et al., Hepatology 2015). Thus, other SRF target genes are also downregulated upon XI-011 treatment.
Minor Comments
Line 78 is confusing, it is not clear what the statistics refer to (mRNA vs genomic copy number)
We have added the missing information (Lines 103 and 531).
Figure 4: What is the ChIP signal normalized to? Are all samples normalized to the same input?
To derive a ΔCt, the samples were normalized to an equal amount of input sample used in each single experiment. Next, the fold enrichment of the specific TF sequence in the ChIP DNA fraction (ΔCt positive) over the negative locus (ΔCt negative-IgG) was calculated (2ΔΔCt, where ΔΔCt = ΔCtpositive – ΔCtnegative). We have added these specifications in the paragraph 4.9 of the Materials and Methods section.
In the introduction it would be worth noting that SRF-VP16 mice develop HCC.
We have added this information to the introduction (Line 92)
Typos
Line 76 – iMDM4 should be “in MDM4”
We have corrected the mistake (now line 102).

Round 2
Reviewer 2 Report
The authors have largely addressed my concerns. They have significantly improved the data around MDM4 inhibition by the small molecule drug XI-011. I believe the authors should be somewhat more cautious with their conclusions with regard to the function of XI-011 and its relationship to MDM4/p53. Specifically, that XI-011 is still highly toxic in a p53ko cell line strongly suggests that its mechanism of action is complex and is not mostly mediated by MDM4 inhibition. Otherwise the authors have notably improved the manuscript which is appropriate for publication.
